# Deubiquitinating Enzyme USP7 Is Required for Self-Renewal and Multipotency of Human Bone Marrow-Derived Mesenchymal Stromal Cells

**DOI:** 10.3390/ijms23158674

**Published:** 2022-08-04

**Authors:** You Ji Kim, Kwang Hwan Park, Kyoung-Mi Lee, Yong-Min Chun, Jin Woo Lee

**Affiliations:** 1Department of Orthopedic Surgery, Yonsei University College of Medicine, 50-1 Yonsei-ro, Seodaemun-gu, Seoul 03722, Korea; 2Brain Korea 21 PLUS Project for Medical Sciences, Yonsei University College of Medicine, 50-1 Yonsei-ro, Seodaemun-gu, Seoul 03722, Korea

**Keywords:** USP7, deubiquitination, self-renewal capacity, multipotential capacity, SOX2, hBMSCs

## Abstract

Ubiquitin-specific protease 7 (USP7) is highly expressed in a variety of malignant tumors. However, the role of USP7 in regulating self-renewal and differentiation of human bone marrow derived mesenchymal stromal cells (hBMSCs) remains unknown. Herein, we report that USP7 regulates self-renewal of hBMSCs and is required during the early stages of osteogenic, adipogenic, and chondrogenic differentiation of hBMSCs. USP7, a deubiquitinating enzyme (DUB), was found to be downregulated during hBMSC differentiation. Furthermore, USP7 is an upstream regulator of the self-renewal regulating proteins SOX2 and NANOG in hBMSCs. Moreover, we observed that SOX2 and NANOG are poly-ubiquitinated and their expression is downregulated in USP7-deficient hBMSCs. Overall, this study showed that USP7 is required for maintaining self-renewal and multipotency in cultured hBMSCs. Targeting USP7 might be a novel strategy to preserve the self-renewal capacity of hBMSCs intended for stem cell therapy.

## 1. Introduction

Human bone marrow-derived mesenchymal stromal cells (hBMSCs) have self-renewal [1,2] and multipotential capacities that allow them to differentiate into osteoblasts, adipocytes, and chondrocytes [3,4,5,6]. hBMSCs have become a valuable resource in regenerative medicine and tissue engineering [7]. hBMSCs express the surface markers CD73, CD90, CD105, but not CD14, CD34, CD45, or human leukocyte antigen-DR [8]. They exhibit plastic adhesion and are usually expanded during passaging [9]. For hBMSCs to be efficiently used in stem cell therapy, a large number of cells have to be cultured over long periods of time in vitro [10]. However, oxidative stress causes hBMSCs to lose their cell proliferation capacity and multipotency during long term cultivation [11]. Oxidative stress affects the lifespan and proliferation of hBMSCs and induces senescence [12]. Recent studies reported that UPPs are essential for the self-renewal of stem cells [13].

Ubiquitin-proteasome pathway (UPP) plays an important role in proteasome-mediated degradation of target proteins [14]. Deubiquitinating enzymes (DUB) can remove the poly-ubiquitin chain to deubiquitinate and stabilize the target protein [15,16]. E3 ligases can ubiquitinate the self-renewal regulatory proteins, thereby promoting stem cell differentiation, whereas DUBs can stabilize them, thus promoting self-renewal and stemness [17]. In hBMSCs, DUBs play a critical role in regulating the stability of self-renewal related proteins such as SOX2, NANOG, and OCT4 [18]. For stem cell therapy, it is essential that hBMSCs maintain their stemness during in vitro culture. Thus, DUB investigation is necessary because DUB-mediated pathways are implicated in regulating stemness of hBMSCs in vitro.

Ubiquitin-specific protease 7 (USP7) is also known as herpesvirus-associated ubiquitin-specific protease (HAUSP) [19,20]. It is involved in various cellular processes, including reaction to DNA damage, transcription, epigenetic regulation of gene expression, immunological response, and viral infection [20]. A previous study showed that USP7 regulates the early differentiation of mouse 3T3-L1 preadipocytes by deubiquitinating the histone acetyltransferase TIP60 [21]. Moreover, USP7 is necessary for osteogenic differentiation of human adipose-derived stem cells (hASCs) [22]. Knockdown of USP7 inhibits chondrogenic differentiation of ATDC5 cells but promotes hypertrophic differentiation [23]. Furthermore, USP7 deubiquitinates and stabilizes the self-renewal related proteins such as SOX2 and NANOG in glioma stem cells (GSCs) [24] and human embryonic stem cells (hESCs) [25]. Previous studies on USP7 primarily focused on its role in tumors [26,27,28,29]. In cancer cells, suppressing USP7 expression increases the degradation of mouse double minute two homolog, an oncogenic E3 ligase, leading to the accumulation of p53, which finally leads to apoptosis [30]. However, the role of USP7 in regulating the proliferation and differentiation of hBMSCs remains unknown. In addition, the expression pattern of USP7 in early and late passage hBMSCs has not been studied yet. This study aimed to investigate the role of USP7 in regulating the self-renewal and differentiation potential of hBMSCs.

## 2. Results

### 2.1. Reduced USP7 Levels Are Associated with Senescence, Decreased Self-Renewal, and Compromised Differentiation Potential of hBMSCs

We first evaluated the proliferation, differentiation, and USP7 levels in early (P4) and late passage (P11) hBMSCs. Late passage hBMSCs were significantly larger and positive for SA-β-gal compared to early passage hBMSCs (Figure 1A). Moreover, P11 cells had a lower proliferation rate (Figure 1B) and colony-forming ability than P4 cells (Figure 1C). The expression of the aging-related markers p53, p21, and p16 was high, while the expression of the self-renewal markers SOX2 and NANOG was low in P11 cells, indicating that the late passage cells were indeed senescent (Figure 1D,E). Furthermore, USP7 mRNA and protein levels were lower in P11 cells compared to P4 cells (Figure 1D,E). When induced to differentiate, P4 cells were more strongly stained with alizarin red stain (ARS) (Figure 1F) and more efficiently differentiated into adipocytes than P11 cells (Figure 1G). Finally, during chondrocyte differentiation, P4 cells formed significantly larger micro-mass pellets than P11 cells (Figure 1H). These results indicate that senescent hBMSCs lose their self-renewal and differentiation capacity, and that USP7 may be involved in regulating these processes.

### 2.2. USP7-Deficient hBMSCs Have a Compromised Self-Renewal Capacity and Multipotency

To investigate the role of USP7 in regulating hBMSC self-renewal and differentiation, siRNA was used to knockdown USP7. The knockdown efficiency of USP7 siRNA was confirmed by Western blot analysis (Figure 2A). The results showed that USP7-deficient hBMSCs had a lower proliferation rate and more reduced colony-forming ability than control cells (Figure 2B,C). Similarly, continuous treatment with the USP7-specific inhibitor GNE6776 reduced the self-renewal capacity of hBMSCs (Appendix A). These results suggest that suppressing USP7 expression or inhibiting USP7 activity decreases the self-renewal ability of hBMSCs. To understand the role of USP7 in hBMSC differentiation, we examined USP7 protein levels during hBMSC trilineage differentiation. hBMSCs were transfected with USP7 siRNA, and osteogenic differentiation was induced for 14 days. ARS staining was then performed, to assess mineralization. USP7 knockdown cells were weakly stained with ARS after osteogenic differentiation (Figure 2D). Moreover, during osteogenic differentiation, we observed a decrease in USP7 protein levels, and USP7 knockdown suppressed osteogenic differentiation (Figure 2E). The expression of osteogenic markers runt-related transcription factor 2 (RUNX2) and osteopontin (OPN) was drastically reduced in USP7 knockdown hBMSCs (Figure 2E). Next, we performed adipogenic differentiation of control and knockdown cells. Oil Red O staining showed that the lipid droplet count was lower in USP7 knockdown cells compared to the control group (Figure 2F). Furthermore, USP7 protein levels decreased steadily during adipogenesis. Knockdown efficacy persisted until the late phase of adipogenic differentiation (Figure 2G). The level of fatty acid-binding protein 4 (FABP4), an adipogenic marker, was lower in the knockdown group compared to the control group (Figure 2G). Finally, chondrogenic differentiation was performed, to evaluate the role of USP7 in chondrogenesis. On day 14 of chondrogenic differentiation, the size of the micro-mass pellet was compared between the control and knockdown groups. The results showed that USP7-deficient cells formed significantly smaller micro-mass pellets than the control cells (Figure 2H). Moreover, USP7 protein levels decreased significantly during chondrogenic differentiation (Figure 2I). The results also showed that the knockdown efficacy was maintained throughout chondrogenic differentiation (Figure 2I). The level of collagen type II alpha 1 chain (COL2a1), a chondrogenic marker, was found to be lower in chondrocytes derived from USP7 knockdown cells compared to that in chondrocytes derived from control cells (Figure 2I). Altogether, these results showed that USP7 knockdown suppresses osteogenic, adipogenic, and chondrogenic differentiation, suggesting that USP7 is essential for hBMSC differentiation.

### 2.3. USP7 Overexpression Enhances Proliferation and Self-Renewal Capability of hBMSCs, but Did Not Affect Multipotency

Our results showed that the USP7 protein levels decreased during hBMSC passaging and differentiation. Therefore, we examined the effect of USP7 overexpression on hBMSC proliferation and differentiation. hBMSCs were transfected with the GFP-tagged lentiviral overexpression vector, and transfection efficacy was determined by Western blotting, which showed that GFP was expressed in transfected cells (Figure 3A). Moreover, USP7 overexpression increased the proliferation rate (Figure 3B) and enhanced the colony forming ability of hBMSCs (Figure 3C), suggesting that USP7 overexpression increases the self-renewal capacity of hBMSCs.

Next, hBMSCs were transfected with USP7 overexpression vector, and osteogenic, adipogenic, and chondrogenic differentiation was induced for 14 days. Differentiation status was analyzed on days 0, 4, 7, and 14. ARS staining and Western blotting revealed no significant differences in osteogenic differentiation capacity between the control and USP7-overexpressing cells on all days analyzed (Figure 3D,E). Similarly, Oil Red O staining and Western blotting revealed no differences in adipogenic differentiation potential between the control and USP7-overexpressing cells (Figure 3F,G). Moreover, the size of the micro-mass pellet derived from the control and USP7-overexpressing hBMSCs was similar, indicating no difference in chondrogenic differentiation potential (Figure 3H). Furthermore, the expression level of the chondrogenic marker SOX9 was nearly identical in both groups on day 14 (Figure 3I). These results indicate that USP7 overexpression has no effect on hBMSC multipotency.

### 2.4. USP7 Interacts with and Stabilizes SOX2 and NANOG

USP7 deubiquitinates and stabilizes the stemness-related proteins, such as SOX2 and NANOG in GSCs [24] and hESCs [25]. Moreover, SOX2-deficient hBMSCs exhibit compromised proliferation and differentiation [31]. We examined whether USP7 stabilizes SOX2 and NANOG in hBMSCs, and whether this regulatory axis affects the self-renewal and multipotency of these cells. To address this, hBMSCs transfected with USP7 siRNA were treated with MG132, a 26S proteasome inhibitor, for 6 h. The results showed that USP7 knockdown decreased the SOX2 and NANOG levels, while MG132 treatment restored the levels of these proteins in USP7 knockdown hBMSCs (Figure 4A). This suggests that USP7 regulates the stability of SOX2 and NONOG at the post-translational level. USP7-GFP overexpression vector and SOX2-Flag or NANOG-EGFP overexpression vectors were co-transfected. The results of immunoprecipitation and Western blotting revealed that USP7 interacts with SOX2 and NANOG (Figure 4B). To examine if USP7 deubiquitinates SOX2 and NANOG in hBMSCs, hBMSCs were co-transfected with siNC or siUSP7, and SOX2 or NANOG or myc-Ub overexpression vectors. Immunoprecipitation and Western blotting revealed that the polyubiquitination of SOX2 and NANOG were increased in USP7-deficient hBMSCs (Figure 4C–E). Altogether, these results indicate that USP7 regulates the self-renewal capacity of hBMSCs by deubiquitinating and stabilizing SOX2 and NANOG.

## 3. Discussion

More than 100 DUBs exist in humans and are categorized into seven classes, based on their sequences and conserved domains [32]. DUBs are critical to the self-renewal and multipotency of MSCs, by causing post-translational modifications (PTM) of self-renewal related proteins [17,33]. The self-renewal related proteins SOX2, NANOG, and OCT4 are required for stem cell self-renewal and multipotency [34]. In hBMSCs, SOX2 knockdown suppresses proliferation and compromises multipotency [31]. USP7 deubiquitinates and stabilizes SOX2 in GSCs to maintain their self-renewal ability and tumorigenic potential [24]. Silencing USP7 decreases the differentiation potential of mouse 3T3-L1 [21], hASCs [22], and ATDC5 cells [23]. However, the role of USP7 in regulating hBMSC self-renewal and differentiation remains unknown.

We found that USP7 is an upstream regulator of the USP7-SOX2 and USP7-NANOG axis in hBMSCs. USP7-deficient hBMSCs showed significantly reduced proliferation and compromised differentiation. Moreover, our data demonstrated that USP7 is required for maintaining the self-renewal capacity and multipotency of hBMSCs. However, whether SOX2- and NANOG-mediated regulation of self-renewal capacity in hBMSCs is dependent on USP7 remains unclear. Nevertheless, our findings are noteworthy, since it was previously recognized that the PTM of SOX2 and NANOG is crucial for regulating self-renewal in MSCs [18,31]. Thus, our finding that USP7 deubiquitinates SOX2 and NANOG, thereby regulating the self-renewal of hBMSCs is novel in the stem cell field. We speculate that regulation of SOX2 and NANOG by USP7 plays a crucial role in the homeostasis of hBMSCs.

USP7 knockdown significantly reduced the osteogenic, adipogenic, and chondrogenic differentiation potential of hBMSCs. Moreover, our USP7 knockdown was highly efficient and lasted till the end of the differentiation period. We speculated that during the early phase of hBMSC differentiation, USP7 might act as a positive regulator. Intriguingly, overexpression of USP7 had no significant effect on hBMSC differentiation. For cells to undergo differentiation, growth arrest is required [35]. However, our results showed that cell proliferation is enhanced following USP7 overexpression. Therefore, cell growth in USP7 overexpressing cells might not have been arrested, which could explain the failure of USP7 overexpression to enhance the differentiation of hBMSCs.

In summary, this study showed that USP7 is a positive regulator of self-renewal and is required for maintaining the multipotency of hBMSCs. Non-clonal cultures of stromal cells contain a subpopulation. Even though no serious adverse effects of non-clonal cultures have been reported thus far, the long-term advantages remain unknown [36]. Nonetheless, due to the limited production of daughter cells in clonal MSC cultures, the use of non-clonal populations in therapy is inevitable [37]. In this study, the validity of hBMSCs was confirmed by flow cytometry, and then subpopulations of hBMSC clones were characterized, as described in our previous studies [38,39]. Additional immunocytochemistry for Ki67, a marker of proliferation, and pRPS6, an active protein synthesis marker, was needed to evaluate the senescent cell population, since the enzyme SA-β-gal is a non-specific marker for cellular senescence [40]. As an alternative, our data provided clues about the reduction of the multipotency and proliferation capacity of senescent hBMSCs. In USP7 overexpressing hBMSCs, the effect of SOX2 and NANOG knockdown on self-renewal needs to be clarified. As we observed that USP7 is significantly downregulated during cellular senescence, further studies are needed to determine whether USP7 overexpression can enhance the self-renewal capacity of late passage hBMSCs. Moreover, the involvement of USP7 in regulating other self-renewal proteins remains unknown. Considering this, the interaction of USP7 with other molecules would be another intriguing subject and worth future investigation. In conclusion, our results suggest that USP7 could be a possible target for preserving the self-renewal capacity in cultured hBMSCs.

## 4. Materials and Methods

### 4.1. Cell Culture

Human bone marrow aspirates were acquired from 3 cm posterior to the anterior superior iliac spine of fifteen adult donors. This research was authorized by the Institutional Review Board of Yonsei University College of Medicine (IRB no. 4-2017-0232). Label-free cell isolation was conducted based on biophysical characterization [41]. hBMSCs were isolated based on adhesion to a plastic culture plate surface, and their validity was verified by flow cytometry, as described in our previous study [38]. Subpopulations of hBMSC clones were characterized, and their multipotency was proven as previously described [39]. Additional hBMSCs were purchased from the American Type Culture Collection (ATCC, Manassas, VA, USA). Cells were cultured as previously described [16]. hBMSCs were cultured in Dulbecco’s Modified Eagle Medium Low-Glucose (DMEM-LG, 31600-034, Gibco, Grand Island, NY, USA) containing 10% fetal bovine serum (FBS, 16000-044, Gibco) and 1% antibiotic-antimycotic solution (15240-062, Gibco). HEK293T cells were purchased from Takara Bio (Mountain View, CA, USA). HEK293T cells were maintained in DMEM High-Glucose (DMEM-HG, 12800-017, Gibco) containing 10% FBS and 1% antibiotic-antimycotic solution (Gibco), according to Takara Bio guidelines. Cells were incubated at 37 °C in a 5% CO_2_ atmosphere.

### 4.2. Senescence-Associated-β-Gal Assay (SA-β-Gal Assay)

Senescence-associated β-gal assay was conducted using a cellular senescence assay kit (Millipore, Temecula, CA, USA), according to the manufacturer’s instructions. Briefly, hBMSCs were seeded in 12-well plates at a density of 1 × 10^4^ cells/well. Cells were then fixed in 1× fixing solution for 10 min at room temperature, washed, and stained with 1× SA-β-gal detection solution for 4 h in the dark at 37 °C. The stained cells were then counted. All experiments were performed in triplicate.

### 4.3. Cell Proliferation Assay

A cell viability assay was conducted using an EZ-Cytox kit (#EZ3000, Daeil Lab, Seoul, Korea). Briefly, hBMSCs were seeded at a density of 1 × 10^4^ cells per well in 12-well plates. Cells were maintained for 7 days in DMEM-LG with 10% FBS. Then, 20 μL of EZ-Cytox solution was added to the culture medium every 2 days, cells were incubated at 37 °C for 3 h, and then placed on to 96-well plates. At 450 nm, absorbance was measured and the medium was then replaced. All experiments were performed in triplicate.

### 4.4. Colony-Forming Unit Fibroblast (CFU-F) Assay

A CFU-F assay was performed as previously described [42]. In brief, hBMSCs were seeded in 10 cm^2^ dishes in amounts of 1 × 10^3^ cells/dish. Then, cells were maintained for 12 days in DMEM-LG supplemented with 20% FBS. The cells were fixed in a 1:1 acetone/methanol fixative and stained for 10 min with a 20% crystal violet solution (Merck, Darmstadt, Germany), and rinsed in distilled water. The cells stained purple were counted. All samples were examined in triplicate.

### 4.5. Osteogenic Differentiation

hBMSCs were seeded in 12-well plates in amounts of 8 × 10^4^ cells/well. The cells were maintained for 14 days in osteogenic medium composed of DMEM-LG containing 10% FBS, 1% antibiotic-antimycotic solution, 10 mM β-glycerophosphate, 100 nM dexamethasone, and 50 μg/mL ascorbic acid. Every 3 days, the osteogenic medium was refreshed. On days 0, 4, 10, and 14, mineralization was determined by Alizarin Red S (ARS) staining. ARS staining was conducted by fixing cells in ice-cold 70% ethanol for 1 h and then washing with water. Then, 1 mL of 3% ARS stock solution was added to each well. The mixtures were allowed to sit for 10 min at room temperature while being gently rotated. Then the cells were washed with water.

### 4.6. Adipogenic Differentiation

hBMSCs were seeded in 12-well plates in amounts of 1 × 10^5^ cells/well and maintained for 14 days in adipogenic medium composed of DMEM-LG containing 10% FBS, 1% antibiotic-antimycotic solution, 0.5 mM isobutylmethylxanthine, 1 μM dexamethasone, 200 μM indomethacin, and 5 μg/mL insulin. Every 3 days, the adipogenic medium was replenished. Oil Red O staining was used to evaluate the adipogenic differentiation capacity. On days 0, 4, 7, and 14, cells were rinsed with PBS and fixed in 3.7% formaldehyde, then 1 mL of 0.18% Oil Red O solution (Sigma-Aldrich, Burlington, MA, USA) was added to each well. The mixtures were left to sit for 30 min at room temperature.

### 4.7. Chondrogenic Differentiation

For chondrogenic differentiation, hBMSCs were cultured as micro-mass. Cells were harvested and dotted in the center of each well of 24-well plates at a density of 1 × 10^5^ cells/10 μL. After allowing the cells to adhere to the surface of the plates for 2 h at 37 °C, chondrogenic differentiation medium composed of DMEM-HG containing 1% antibiotic-antimycotic solution, 1% Insulin Transferrin Selenium-A (ITS) (Invitrogen, Carlsbad, CA, USA), 1 μM dexamethasone, 10 μM L-proline, 50 μg/mL ascorbic acid (Invitrogen), and 10 ng/mL TGF-β3 (R&D Systems, Minneapolis, MN, USA) was gently added, and the culture was maintained for 14 days. The chondrogenic differentiation medium was replenished every 2 days. Micro-mass pellets were then fixed in 4% paraformaldehyde for 2 days and then stained with alcian blue to detect proteoglycans.

### 4.8. Western Blotting

Total protein was isolated from cells using Pro-Prep protein extraction solution (Intron, Seongnam, Korea). Next, 30 μg of protein was extracted from each sample, and the protein was separated using sodium dodecyl sulfate-polyacrylamide gel electrophoresis (SDS-PAGE) and transferred to polyvinylidene difluoride (PVDF) membranes. Membranes were blocked with 5% skim milk (BD, Sparks, MD, USA) in tris-buffered saline with Tween 20 (TBST). The membranes were then incubated with primary antibodies overnight at 4 °C. After incubation, membranes were washed three times with TBST (10 min each) and then incubated with secondary antibodies for 1 h at room temperature. Finally, all membranes were washed three times with TBST (10 min each) before imaging with either a LAS 4000 X (Fujifilm, Tokyo, Japan), Amersham ImageQuant800 biomolecular imager (Cytiva, MA, USA) or a C-DiGit Blot Scanner (LI-COR, Lincoln, NE, USA). Details of primary antibodies are listed in Appendix A.

### 4.9. Quantitative Real-Time-Polymerase Chain Reaction (qRT-PCR)

Total RNA was extracted from cells using an AccuPrep Universal RNA Extraction Kit (Bioneer, Korea), following the manufacturer’s instructions. Then, 2 μg total RNA was reverse transcribed using an Omniscript kit #205113 (Qiagen, Germantown, MD, USA). A Step One Plus Real-time PCR System (Applied Biosystems, Foster City, CA, USA) and 2× qPCRBIO SyGreen Mix #PB20.12-05 (PCR Biosystems, London, UK) were used to perform quantitative real-time polymerase chain reactions (qRT-PCR). All primer sequences are listed in Appendix A.

### 4.10. Plasmids and siRNAs

hBMSCs were seeded in 6-well culture plates in amounts of 1 × 10^5^ cells/well, and the cells were transfected with USP7 targeting siRNA or USP7 lentiviral expression vectors using the X-tremeGENE 9 DNA Transfection Reagent (Roche, Mannheim, Germany), according to the manufacturer’s instructions. After 6 h of transfection, the medium was replenished with DMEM-LG supplemented with 10% FBS. Scrambled negative controls (siRNA no. SN1002) and USP7 siRNA (siRNA no. 7874-1) were purchased from Bioneer (Daejeon, Korea). Lentiviral control vector (pLenti-GIII-CMV-GFP-2A-Puro, LV590) was purchased from Applied Biological Materials (Viking Way Richmond, BC, Canada). GFP-USP7 (pLenti-GIII-CMV-GFP-USP7-2A-Puro, NM003470) and FLAG-SOX2 vectors (pBabe puro DEST Flag SOX2, #45270) were purchased from Addgene (Watertown, MA, USA). EGFP-NANOG expression vector (pEGFP-C1-NANOG) was also used. Myc-Ub vector was kindly provided by Prof. Jae-Hyuck Shim (University of Massachusetts Medical School, Worcester, MA, USA).

### 4.11. Immunoprecipitation Analysis

Transfected cells were treated with 10 μM MG132 (cat. no. s2619, Selleckchem) for 6 h. The cells were then lysed using a non-denaturing lysis buffer [20 mM Tris-HCl (pH 8.0), 137 mM NaCl, 0.5% Nonidet P40, 2 mM EDTA (pH 8.0), 1 mM phenylmethylsulfonyl fluoride, phosphatase inhibitor (P3200-001, GenDepot), and protease inhibitor (P3100-001, GenDepot)]. Lysates were incubated with relevant antibodies (anti-USP7, anti-FLAG, anti-NANOG, anti-ubiquitin, or IgG) overnight at 4 °C, with moderate rotation. After incubation, the lysates were washed and conjugated with protein A/G agarose beads (sc-2003, Santa Cruz Biotechnology) for 4 h at 4 °C. Total cell lysates (input) and immunoprecipitates were separated by SDS-PAGE, transferred to PVDF membranes, then immunoblotted with the indicated primary antibodies. 2× Lane Marker Reducing Sample Buffer (#39000, Thermo Fisher Scientific, Rockford, IL, USA) was used to elute the immunoprecipitates. Details of primary antibodies are listed in Appendix A.

### 4.12. Statistical Analysis

All experiments were performed at least thrice. Statistical significance was set at *p* < 0.05. All data are expressed as mean ± standard deviation (SD). Student’s *t*-test was used to assess statistically significant differences between groups.

## Figures and Tables

**Figure 1 ijms-23-08674-f001:**
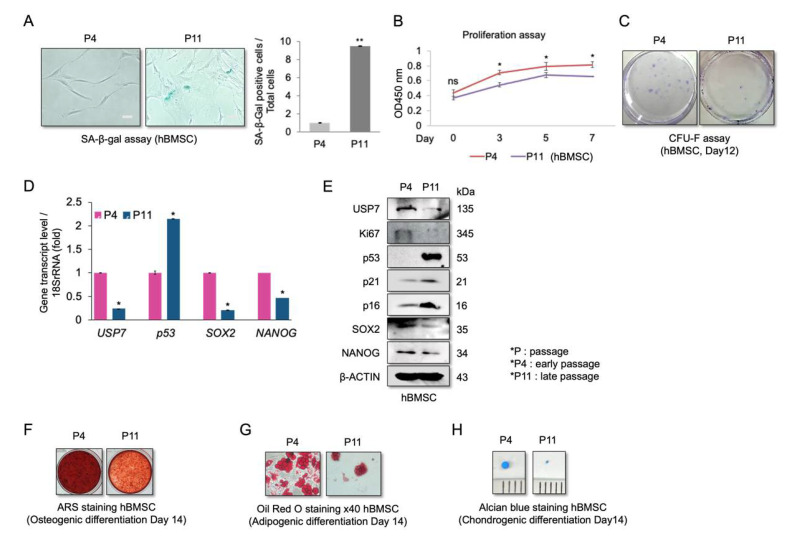
Analysis of self-renewal, multilineage differentiation potential, and USP7 expression in early and late passage hBMSCs. (**A**) Senescence-associated-β-gal (SA-β-gal) activity investigated in passage 4 (P4) and passage 11 (P11) hBMSCs. Scale bar = 30 μm. (**B**) Proliferation rates of P4 and P11 cells measured using the cell viability assay. (**C**) Colony-forming capacities of P4 and P11 cells measured using CFU-F. (**D**) qRT-PCR and (**E**) immunoblot analysis results for USP7, proliferation marker Ki67, aging markers p53, p21, and p16, and self-renewal markers SOX2 and NANOG. (**F**) Alizarin red S stained P4 and P11 cells after 14 days of osteogenic differentiations. (**G**) Oil red O stained P4 and P11 cells cultured in adipogenic medium for 14 days. Scale bars for the original magnification × 40 = 30 μm. (**H**) Alcian blue stained P4 and P11 micro-mass pellets cultured in chondrogenic medium for 14 days. Data are presented as mean ± SD. ns; not significant, * *p* < 0.05, ** *p* < 0.01. *n* = 3.

**Figure 2 ijms-23-08674-f002:**
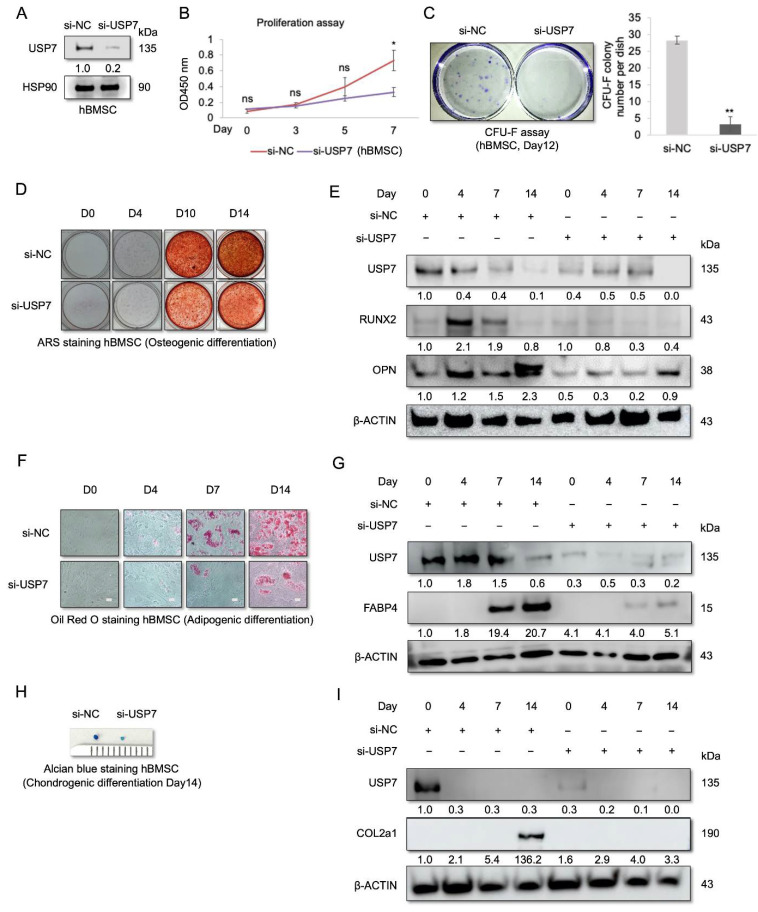
Effect of USP7 knockdown on the self-renewal and differentiation of early passage hBMSCs. (**A**) Immunoblot analysis of hBMSCs following transfection with the negative control or USP7 siRNA. (**B**) Results of the cell proliferation assay. (**C**) Results of the colony-forming unit fibroblast (CFU-F) assay. On day 12 of the CFU-F assay, colonies were stained with crystal violet. (**D**) Results of the ARS staining. (**E**) Representative immunoblots depicting results of osteogenic differentiation. (**F**) Results of the Oil Red O staining. Scale bar = 200 μm. (**G**) Representative immunoblots depicting results of adipogenic differentiation. (**H**) Alcian blue staining results. (**I**) Representative immunoblots depicting results of chondrogenic differentiation. Data are presented as mean ± SD. ns; not significant, * *p* < 0.05, ** *p* < 0.01. *n* = 3.

**Figure 3 ijms-23-08674-f003:**
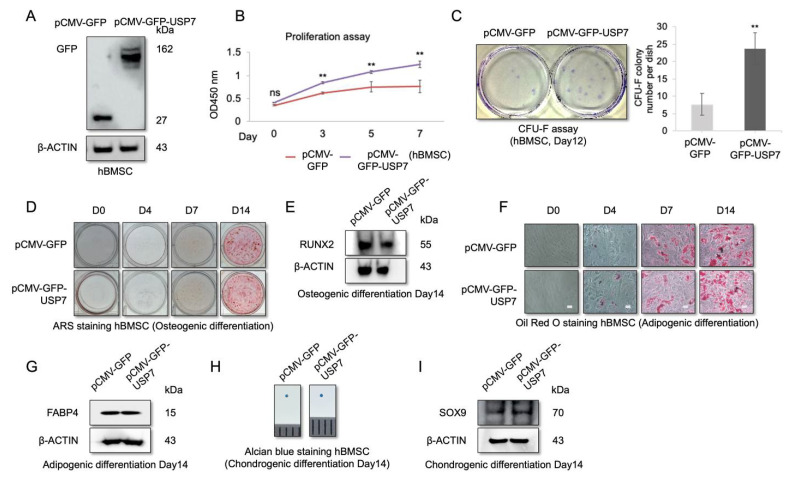
Effect of USP7 overexpression on self-renewal and differentiation of hBMSCs. (**A**) Immunoblot depicting the results of hBMSCs transfected with pCMV-GFP and pCMV-GFP-USP7 vector. (**B**) Cell proliferation analysis results. (**C**) CFU-F assay results. (**D**) ARS staining results. (**E**) Immunoblot analysis of osteogenic markers. (**F**) Oil Red O staining results. Scale bar = 200 μm. (**G**) Immunoblot analysis of adipogenic markers. (**H**) Alcian blue staining results. (**I**) Immunoblot analysis of chondrogenic markers. Data are presented as mean ± SD. ** *p* < 0.01. *n* = 3.

**Figure 4 ijms-23-08674-f004:**
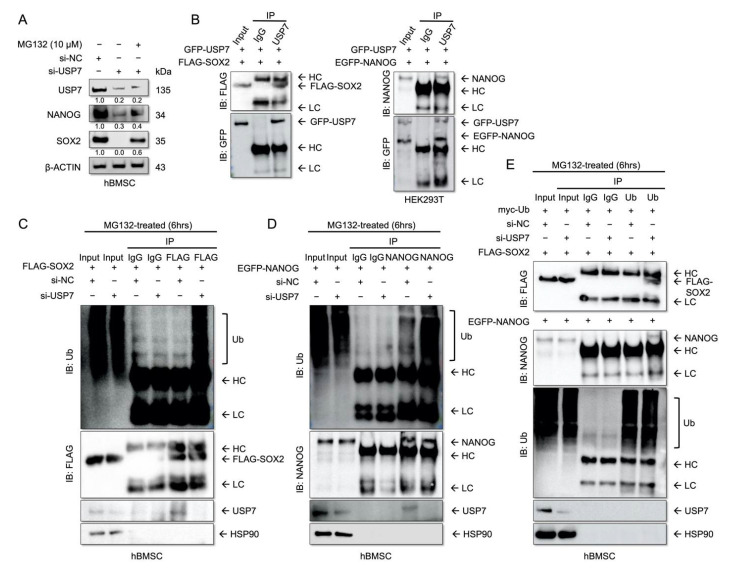
USP7 deubiquitinates and stabilizes the self-renewal-related proteins. (**A**) Immunoblot analysis of cell lysates. (**B**) Results of the immunoprecipitation assay. hBMSCs were co-transfected with the indicated plasmids for 48 h and then treated with 10 μM MG132 for 6 h. (**C**) hBMSCs were co-transfected with the indicated siRNA and pPURO-FLAG-SOX2 for 48 h and then treated with 10 μM MG132 for 6 h. Cell lysates were subjected to immunoprecipitation with anti-FLAG antibody, and immunoblotting was performed using anti-Ubiquitin (Ub) antibody. (**D**) hBMSCs were co-transfected with the indicated siRNA and pEGFP-NANOG for 48 h and then treated with 10 μM MG132 for 6 h. Cell lysates were subjected to immunoprecipitation with anti-NANOG antibody, and immunoblotting was performed with anti-Ub antibody. (**E**) hBMSCs were co-transfected with the indicated plasmids for 48 h and then treated with 10 μM MG132 for 6 h. Cell lysates were subjected to immunoprecipitation with anti-USP7 antibody, and immunoblotting was performed with anti-FLAG and anti-NANOG antibodies. HC, heavy chain; LC, light chain. *n* = 3.

## Data Availability

All data generated in this study can be obtained from the corresponding author upon reasonable request.

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
