# Peer review of "Deubiquitinating Enzyme USP7 Is Required for Self-Renewal and Multipotency of Human Bone Marrow-Derived Mesenchymal Stromal Cells"

_ijms, 2022, doi:10.3390/ijms23158674_

Round 1

Reviewer 1 Report

In general, I read good research work, with many experiments and organized and related each other. The paper is well written and nice presented. Aim and scopes of the journal are respected. I have only few comments.

1.     HEK9237 is not clear. They are not mentioned in Material&method and are written in Fig. 4B. Is that an error?

2.     Why si-USP17 is not completely efficient, as considered visible bands in Figure 2E-G and Figure 4a?

3.     The authors should think and inform trhough the eventual publication of such their manuscript that other important and new methods to select stem cells from bone marrow, for example label-free biophysical characterizazion, are existing. I suggest to read the article located at doi.org/10.3390/bioengineering9020049 Zia et al. 2022 and very briefly mention this concept.

Reviewer 2 Report

In this study, the author investigated the role of USP7 in regulating the self-renewal and differentiation potential of human bone marrow-derived mesenchymal stem cells (hBMSCs). They found that USP7 is a positive regulator of self-renewal and is required for maintaining the multipotency of hBMSCs.

This work is not very innovative because it has already been reported that: USP7 is an essential player in the osteogenic differentiation of hASCs (PMID: 28807012); USP7 interacts with and stabilizes REST by preventing SCFβ-TrCP-mediated ubiquitination, thus promoting the maintenance of stemness (PMID: 27285106); USP7 regulates female germline stem cell self-renewal through DNA Methylation (PMID: 33151468). The innovation in this work is that the authors evaluated the role of USP7 in early and late passage hBMSCs, which, in any case, turned out to be very similar to AMSCs due to their origin.

Before publication, the authors have to indicate a detailed and proper definition of human bone marrow-derived mesenchymal stromal cells (hBMSCs) in the “Introduction” paragraph. They define these cells as stem cells, but the proper definition is stromal cells. The authors should also better explain that the isolation of hMESCs, according to current criteria, produces heterogeneous, non-clonal cultures of stromal cells containing stem cells with different multipotential properties, committed progenitors and differentiated cells (PMID: 26423725). Furthermore, they should report how they characterized the hBMSCs by evaluating the surface markers expression and demonstrating the ability of the hMESCs to differentiate into osteoblasts, adipocytes and chondroblasts in vitro as reported by Dominici et al. (PMID: 16923606).

To correctly evaluate the presence of senescence cells in the cell cultures, the authors could examine the expression of Ki67, a marker of cycling cells and not of resting cells, pRPS6, an active protein synthesis marker that has been proposed to allow distinction between senescent and quiescent cells, along with the evaluation of senescence-associated beta-galactosidase activity. The Ki67 (−) pRPS6 (+) SA-β-gal (+) cells represent senescent cells (PMID: 33803589). The reliability of the associated-beta-galactosidase activity is questionable because the enzyme is a non-specific marker for cell senescence. In addition, the authors can evaluate changes in cellular senescence by western blot analysis of the levels of expression of various proteins involved in senescence and cell cycle exit, not only p53 but also other proteins such as RB - RB2 - p107 - p27 -p21 - ARF - p16.

In Figure 1 legend there is an error in the caption of letters D and E that must be reversed.

For the western blotting experiments reported in Figure 2, it might be useful to add quantization of the experiments to improve and increase understanding of the data.

Line 155 of the paper cannot be read due to the overlapping of the figure.

Round 2

Reviewer 2 Report

Authors addressed my concerns.